# Active Learning for Efficient Discovery of Optimal Combinatorial Perturbations

Jason Qin [1]   Hans-Hermann Wessels [1]   Carlos Fernandez-Granda [2 3]   Yuhan Hao [1]

## Abstract

Combinatorial CRISPR screening enables large-scale identification of synergistic gene pairs for combination therapies, but exhaustive experimentation is infeasible. We introduce **NAIAD**[1], an active learning framework that efficiently discovers optimal gene pairs by leveraging single-gene perturbation effects and adaptive gene embeddings that scale with the training data size, mitigating overfitting in small-sample learning while capturing complex gene interactions as more data is collected. Evaluated on four CRISPR datasets with over 350,000 interactions, NAIAD trained on small datasets outperforms existing models by up to 40%. Its recommendation system prioritizes gene pairs with maximum predicted effects, accelerating discovery with fewer experiments. We also extend NAIAD to optimal drug combination identification among 2,000 candidates. Overall, NAIAD enhances combinatorial perturbation design and drives advances in genomics research and therapeutic development in combination therapy. Our code is publicly available at: https://github.com/NeptuneBio/NAIAD

## 1. Introduction

Targeting multiple genes through drug combinations or polypharmacology offers a transformative therapeutic approach for developing effective treatments across diverse medical fields, including oncology (Al-Lazikani et al., 2012; Mokhtari et al., 2017), infectious diseases (Hammond et al., 2022; Shyr et al., 2021), and metabolic disorders (Samms et al., 2020; Jastreboff et al., 2023). Combinatorial gene perturbations can yield additive or synergistic effects, enhancing therapeutic outcomes beyond what is achievable with single-gene targeting (Nature Medicine, 2017; Hwangbo et al., 2023). One of the most notable successes in transitioning cellular phenotype is the discovery of the Yamanaka factors—a specific combination of four transcription factors capable of reprogramming differentiated cells back to a pluripotent state (Takahashi & Yamanaka, 2006). This groundbreaking achievement demonstrates the significant potential within the combinatorial perturbation space to engineer cellular phenotypes.

The critical question now is how to systematically identify additional effective combinatorial perturbations that can transform cells to achieve desired phenotypes. Comprehensively exploring this huge space presents a mathematical challenge due to the exponential growth of possible combinations. With approximately 20,000 protein-coding genes in the human genome, the total number of two-gene combinations approaches 200 million, and for four-gene combinations exceeds 6 quadrillion ($10^{15}$). Experimentally testing all possible combinations is infeasible. Therefore, developing computational models that can predict the most effective gene combinations is essential for the efficient identification of the most effective combinatorial perturbations. Notably, active learning frameworks (Eisenstein, 2020), such as the AI + Experiment Loop (Rood et al., 2024), have offered promising solutions by enabling efficient exploration of this space.

We frame the discovery of optimal gene or drug combinations as a machine learning problem of active search over a high-dimensional combinatorial space, where evaluating each combination (via experiment) is costly. Our method trains a neural surrogate model that predicts the effects of unseen perturbation pairs by combining overparameterized encodings of single-gene outcomes with a learned gene embedding space that models interaction effects. The surrogate guides new experiment selection via acquisition strategies inspired by Bayesian optimization, with the ability to leverage both exploitation and exploration.

While this work is motivated by biological discovery, similar challenges arise across machine learning domains—including data augmentation policy search in vision (Cubuk et al., 2019), cold-start item selection in recommender systems (De Pessemier et al., 2021), and sample-efficient policy learning in robotics (Anwar et al., 2025)—all

---

[1]Neptune Bio, New York, NY, USA [2]Center for Data Science, New York University, New York, NY, USA [3]Courant Institute of Mathematical Sciences, New York University, New York, NY, USA . Correspondence to: Yuhan Hao <yuhan@neptune.bio>.

*Proceedings of the 42nd International Conference on Machine Learning*, Vancouver, Canada. PMLR 267, 2025. Copyright 2025 by the author(s).

[1]Named after the innermost satellite of the planet Neptune

of which involve large discrete spaces, costly evaluations, and the need for adaptive modeling and decision making. In such domains, discrete components (e.g., transformations, items, actions) play a role analogous to gene or drug perturbations in our framework: each is embedded in a latent space, and combinations of these embeddings are used to represent and evaluate complex configurations. Our framework shows how these components can be actively selected via a data-adaptive surrogate to enable efficient, scalable discovery.

In this work, we introduce NAIAD, an active learning framework for identifying the most effective gene or drug combinations. We initially train a model on a small dataset from experiments, which enables it to predict unseen combinatorial perturbation effects across the entire combinatorial space. These predictions guide the design of subsequent CRISPR screening libraries for targeted experiments, thus allowing us to iteratively refine the model and converge on optimal combinations (Figure 1A). We focus on modeling the effects of 2-gene combinations, combining knowledge of single-gene effects with predictions of gene-gene interactions (Figure 1B). Our key contributions are:

(1) A novel combinatorial perturbation model which incorporates adaptive gene embeddings that scale with the training data size, along with an overparametrized representation of single-gene perturbation effects.

(2) Maximum Predicted Effect (MPE)-based recommendation system that suggests gene combinations for subsequent CRISPR library design, facilitating the discovery of synergistic and effective gene combinations.

(3) An AI + Lab active learning framework that effectively identifies optimal gene combinations, significantly reducing the number of experimental iterations needed to achieve robust results.

## 2. Related Work

CRISPR combinatorial perturbation technologies can be broadly classified into two main categories (Norman et al., 2019): single-cell combinatorial perturbation and bulk combinatorial perturbation. Single-cell combinatorial perturbation measures the entire transcriptome, capturing comprehensive gene expression changes in individual cells, but with a limited number of gene combinations. In contrast, bulk combinatorial perturbation focuses on measuring a single phenotype, enabling the investigation of a much broader range of gene combinations.

Predicting combinatorial perturbations has been a significant challenge due to non-linearity of certain gene combinations. Various machine learning approaches have been proposed to address this problem using single-cell combinatorial per-

turbation data. Variational Autoencoders (VAEs) (Kingma & Welling, 2014) have been employed to model genetic and chemical combinatorial perturbations by simultaneously learning embeddings of single perturbations and capturing non-linear interactions in methods such as CPA (Lotfollahi et al., 2023), Com$\beta$VAE (Geiger-Schuller et al., 2023), sVAE+ (Lopez et al., 2023), and SAMS-VAE (Bereket & Karaletsos, 2023). These approaches facilitate the modeling of complex relationships between genes or compounds within the latent space of embeddings. Methods such as sVAE+ (Lopez et al., 2023) and SAMS-VAE (Bereket & Karaletsos, 2023) have been developed to model sparsity in latent variable intervention effects. By disentangling the perturbation-related sparse latent space, these models effectively identify critical features and interactions within high-dimensional biological data. SALT&PEPER implements a method to separately learn linear and non-linear effects of gene perturbations (Gaudelet et al., 2024). By using gene-embedding-based autoencoder models, they effectively decompose the interactions, enabling more interpretable models of gene effects. Recently, several single-cell foundation models, such as scGPT (Cui et al., 2024) and scFoundation (Hao et al., 2024), trained on all publicly available observational data, have demonstrated their ability to predict cellular responses following perturbations after model fine tuning. Additionally, GEARS leverages graph neural networks (GNNs) (Kipf & Welling, 2017) to incorporate prior biological knowledge into the network architecture (Roohani et al., 2023). GNNs facilitate the inference of gene-gene interactions by leveraging known pathways and interaction networks, thus enhancing the prediction of combinatorial effects.

However, most of these methods primarily focus on predicting post-perturbation gene expression profiles and do not extend to the prediction of phenotypic outcomes. Also, the number of gene combinations from single-cell combinatorial perturbations dataset is very limited (100-200 gene combinations), so it is difficult to evaluate if those deep learning models are generalizable in the entire combinatorial perturbation space and can outperform linear models (Ahlmann-Eltze et al., 2024).

Some existing methods (i.e. GEARS (Roohani et al., 2023) and CPA (Lotfollahi et al., 2023)), are capable of predicting combinatorial perturbations from single-cell transcriptomic profiles, as well as from single measurements derived from bulk screens (i.e. cell viability). However, these approaches generally assume the availability of sufficient data that would allow for a substantial portion to be used for training. In practice, we are often limited by the number of training samples due to the exponential growth of possible combinations in combinatorial perturbation data. This limitation necessitates having methods that can perform well with minimal data. Active learning frameworks offer a solu-

tion by optimizing model performance while using the least amount of training data possible. RECOVER utilizes an active learning framework that iteratively selects the most promising drug combinations for testing through the AI + Lap loop (Bertin et al., 2023). RECOVER applies a bilinear operator to create permutation-invariant representations that integrate multiple single perturbation effects, learning the non-linear components of drug combinations. It leverages ensemble models to estimate uncertainty in the predictions through deep ensembles (Lakshminarayanan et al., 2017), guiding the selection of experiments for subsequent rounds based on both predicted effects and associated uncertainties.

## 3. Methods

In this section, we present the architecture of our model for predicting the effects of combinatorial perturbations, along with our AI + Lab active learning framework (Figure 1A). Our NAIAD framework addresses two key objectives: (1) achieving predictive accuracy with limited training data from the initial experimental round; and (2) implementing a recommendation strategy to select additional gene pairs that maximize information gain, thereby accelerating convergence with fewer AI + experimental iterations. Ultimately, our goal is to optimize the use of limited experimental resources, reducing the need for exhaustive testing of all possible combinations and efficiently identifying effective gene combinations that drive cells toward desired phenotypes.

### 3.1. Model Design

Let $X^{\text{gene}} \in \mathbb{R}^{k \times p}$ be the learnable gene embedding matrix, where $k$ is the number of unique genes perturbed within the dataset and $p$ is the dimension of the gene embeddings, which is adapted to the number of training samples. Let $Y_i$ be a scalar value representing the effects of a single-gene ($i$) perturbation. In this work, we focus on the case of perturbing two genes simultaneously, so the value $Y_{i+j}$ denotes the combined effect of the two-gene ($i + j$) perturbation. The target variable $Y$ is a scalar that can represent various biological outcomes, such as cell fitness, marker enrichment levels, or projected gene signatures derived from single-cell transcriptomic data.

Our model is formulated as:

$$Y_{i+j} = \phi([Y_i, Y_j]W_1)A_1^T \\ + f(\phi(W_2 X_i^{\text{gene}}), \phi(W_2 X_j^{\text{gene}}))A_2^T \quad (1)$$

where $\phi$ is an activation function (ReLU or GeLU), and $W, A$ are learnable matrix parameters. $X_i^{\text{gene}}$ and $X_j^{\text{gene}}$ are the row-vectors for gene $i$ and $j$ in $X^{\text{gene}}$. $\phi([Y_i, Y_j]W_1)A_1^T$ models the over-parameterized single-gene effects, and $f(\phi(W_2 X_i^{\text{gene}}), \phi(W_2 X_j^{\text{gene}}))A_2^T$ models the genetic inter-

action contributions between $i$ and $j$, where $f$ applies a permutation-invariant function to capture the interactions of genes $i$ and $j$ through their embeddings $X^{\text{gene}}$.

By explicitly incorporating single-gene effects as input parameters, we condition combinatorial predictions on the known single-gene effects, which substantially enhances the model's performance. The parameter $W_1$ acts as an over-parameterized encoder, projecting the data from $k$ dimensions to a much higher-dimensional space $\mathbb{R}^m$, where $m \gg k$. The projection of this data into a high-dimensional (though finite) space allows us to capture intricate patterns that are not discernible in the original low-dimensional space.

To model genetic interactions, we sum the gene embeddings of the perturbed gene combinations in the latent space to obtain a single combined embedding. This summation is a permutation-invariant operation, and thus independent of the order of the multiple perturbations. The combined embedding is then passed through an encoder that compresses it into a singular interaction value. This compression reduces the dimensionality of the data while retaining the essential information needed to predict nonlinear interaction of genes. Finally, the compressed representations from both the over-parameterized single-gene effect and combined-gene embeddings are used to predict the phenotype. This also allows our model to learn higher-order interactions among gene combination perturbations, leading to improved predictive performance.

We can interpret the model as follows: the overparameterized single-gene perturbation effects tell the model how any pair of genes additively would impact the cellular phenotype, and relationships between gene embeddings predict synergistic or buffering interactions between gene pairs.

Additionally, while we focus on 2-gene perturbations in this study, our model can naturally be extended for sets of $q > 2$ simultaneously perturbed genes. For a set involving $S = \{i_1, i_2, \ldots, i_q\}$ perturbations, we can model the effect of jointly perturbing the genes in $S$ as:

$$Y_S = \phi\left([Y_{i_1}, Y_{i_2}, \ldots, Y_{i_q}]W_1\right)A_1^T \\ + \left(\sum_{i \in S} \phi\left(W_2 X_i^{\text{gene}}\right)\right)A_2^T \quad (2)$$

where we use summation as the permutation-invariant function that combines the embeddings of individual perturbed genes.

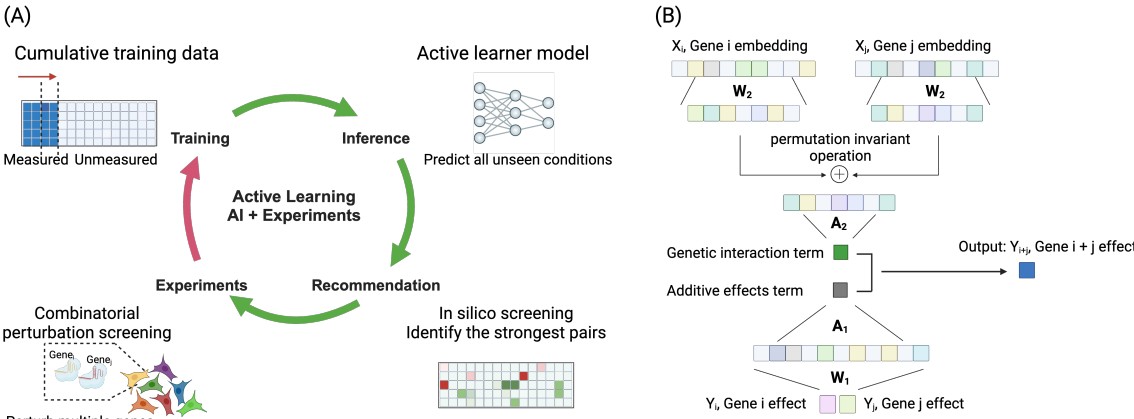

*Figure 1.* Illustration of active learning framework in CRISPR combinatorial perturbation (A) and our NAIAD model architecture with overparameterized single-gene effects and adaptive gene embedding modules (B).

## 3.2. Small Sample Learning as an Important Initial Step for Active Learning

Unlike most existing models, we propose modeling gene embeddings within a latent space with varying dimensionality tailored to the training-data size. Training large models on small datasets can lead to overfitting. To mitigate this, we initialize the dimension $p$ of the latent space to a small value at the beginning of training. As active learning iteratively incorporates new data, the size of the training dataset continually increases, and we correspondingly increase $p$ following a predetermined schedule based on the average number of times each gene is seen in the dataset. As the training dataset expands, the increased dimensionality of the latent space enables the model to capture more complex patterns and interactions among genes without overfitting to noisy experimental measurements. By controlling the model complexity based on the available data, this adaptability allows the gene embeddings to effectively leverage both small and large training datasets, optimizing model performance across different data sizes.

## 3.3. Recommendation System of NAIAD

A critical component of our active learning framework is the recommendation system for selecting gene combinations in subsequent experiment rounds to acquire new data (Figure 1A). We evaluated an ensemble-based uncertainty estimation, where multiple models with different initializations are aggregated to estimate prediction uncertainty through measures like variance and entropy (Lakshminarayanan et al., 2017). This ensemble approach not only improves predictive performance and enhances interpretability, but also quantifies epistemic uncertainty arising from limited data. We investigated uncertainty calculation by maximizing the likelihood $P(Y|\hat{\mu}, \hat{\sigma})$ under the assumption of a conditional Gaussian distribution (Lahlou et al., 2023) as well. How-

ever, we noted that the experimental uncertainty estimated from the ensemble method was more stable. Thus, in this work, we adopted the variance of ensemble predictions as the ensemble-based uncertainty estimator.

In addition to sampling gene pairs with high uncertainty, we incorporate maximum-predicted-effect (MPE) and residual-based sampling strategies to diversify our experimental selection. The MPE sampling focuses on gene pairs with strong predicted effects, identifying combinations that may yield substantial biological insights or therapeutic benefits. Additionally, to balance the exploitation of known MPE areas with the exploration of uncertain regions, we also combined ensemble prediction uncertainty with residual-based sampling. Residual-based sampling targets areas where the model's predictions deviate most strongly from a linear model baseline, allowing exploration of complex interactions that the model has not yet captured and helping to uncover gene interactions that might be missed by linear models. Combining residual-based sampling with uncertainty estimation is equivalent to the Upper Confidence Bound (UCB) sampling method used in RECOVER.

# 4. Experiments

## 4.1. Datasets

We evaluate our models on cell-viability measurements across two cell types from four bulk combinatorial CRISPR perturbation screening datasets and one drug combination screening dataset (Norman et al., 2019; Simpson et al., 2023; Horlbeck et al., 2018; Zheng et al., 2021; Bertin et al., 2023). We treat each gene combination or drug combination as one sample. Each dataset contains symmetric, pair-wise measurements for its gene or drug combinations, allowing us to comprehensively compare model predictions with ground truth measurements. Detailed descriptions of these datasets

are provided in Appendix A.

## 4.2. Downsampling Experiments

We evaluate model performance on different amounts of training data. Since our datasets vary in number of features and overall size, we split the data differently depending on the experiment and dataset used. In our downsampling experiments in Section 5.2, we used [100, 200, 350, 500, 750, 1000, 1250, and 1500] samples during training for the Norman et al. (2019) dataset (6,328 combinations), and [100, 500, 1000, 2000, 3000, 4000, 5000, 6000] samples for training on the Simpson et al. (2023) (147,658 combinations) and Horlbeck et al. (2018) datasets (100,576 combinations for K562; 95,703 combinations for Jurkat T), along with 10% and 30% of each dataset for validation and testing, respectively.

## 4.3. Active Learning Experiments

To mimic an active learning scenario, in Section 5.3 we started with 100 samples from the Norman dataset in the first round, and incrementally included an additional 100 samples each active learning round for 4 additional rounds. For the Simpson and two Horlbeck datasets, we began with 500 samples in the first round, and incrementally included 500 more each round for 4 additional rounds.

The data for the first round is selected uniformly, and the incremental data added in each subsequent round is selected via an acquisition function (see Appendix C for description of different acquisition functions used). The iterative data selection process allowed us to progressively improve the models by incorporating more data based on the active learning strategy.

In Section 5.4, we extend our active learning framework to a different intervention modality - drug treatment. We use the Bertin et al. (2023) drug dataset, and follow the same sampling schedule as for the Norman gene dataset.

## 5. Results

### 5.1. Small Sample Learning and Adaptive Gene Embeddings

One of the primary challenges in active learning is that it typically begins with a small training dataset (Chandra et al., 2020). When the amount of training data is limited, linear models consistently outperform deep learning models. We observed similar patterns in this study particularly when the training data size is below a certain threshold (e.g., 10% in the (Norman et al., 2019) dataset, corresponding to 20 observations per gene during training). As the training data size increases beyond this threshold, the Multi-Layer Perceptron (MLP) begins to surpass the linear model in

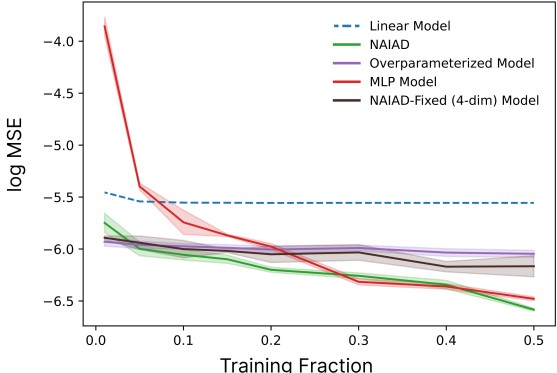

*Figure 2.* Performance on test data of different gene embedding settings in NAIAD using the Norman dataset (4,429 training combinations) across varying training data sizes, reported as $\log(\text{MSE})$. The models without gene embeddings or with low-dimensional embeddings perform well with small training data but do not improve as more data are added. In contrast, the MLP model with larger gene embeddings outperforms these models when the training data exceeds 30%. The adaptive embedding approach achieves the best performance across all training data sizes.

performance (Figure 2).

We investigated different gene embedding configurations: an over-parameterized single-gene effect model without embeddings, NAIAD with fixed 4-dimensional gene embeddings, and NAIAD with training-data-size adaptive embeddings ranging from 2 to 128 dimensions. The results suggest that adaptive embeddings consistently achieve superior performance in most cases, regardless of the training data size (Figure 2) in the Norman dataset. This adaptability allows the model to effectively leverage the strengths of both embedding representations and combined single-gene effects. Of note, we observed that the compressed scalar values from the single-gene components strongly correlate with the linear-model predicted results, demonstrating that the NAIAD model leverages the strong baseline performance of linear models. Moreover, the correlation between the scalar values from the gene embedding components and the linear residuals becomes stronger as the training data size increases (Appendix Figure 6).

### 5.2. Benchmark Analysis in Small Sample Learning

We benchmarked NAIAD against the linear model, MLP, GEARS, and RECOVER on the previously-described four bulk combinatorial perturbation datasets that cover 6,328; 147,658; 100,576; and 95,703 gene combinations (detailed information of benchmark models is described in Appendix B). NAIAD consistently outperformed all other models, particularly in situations with a limited number of observed gene pairs, as measured by $\log(\text{Mean Square Error})$, Pear-

*Table 1.* Root mean square error (RMSE) of five models on test data when each gene is observed in training data approximately 4 and 20 times. Error is standard error (SE) across three cross-fold replicates.

| Gene Frequency | Model | Dataset RMSE ($\times 10^{-2}$) | | | |
| --- | --- | --- | --- | --- | --- |
| | | Norman | Simpson | Horlbeck K562 | Horlbeck Jurkat |
| 4 | Linear | 6.2 (1.7) | 3.3 (0.3) | 6.4 (0.9) | 3.9 (0.6) |
| | MLP | 7.7 (0.9) | 4.3 (0.9) | 7.8 (2.5) | 5.5 (2.1) |
| | GEARS | 16.6 (19.8) | 5.4 (3.4) | 13.0 (11.6) | 13.5 (16.1) |
| | RECOVER | 7.1 (2.8) | 3.9 (0.4) | 7.9 (2.0) | 5.0 (0.7) |
| | NAIAD | **5.1 (1.8)** | **2.2 (0.1)** | **6.1 (1.9)** | **3.0 (0.6)** |
| 20 | Linear | 6.1 (1.1) | 3.3 (0.2) | 6.4 (0.9) | 3.8 (0.6) |
| | MLP | 5.0 (1.4) | 2.0 (0.3) | 5.9 (0.1) | 3.0 (0.4) |
| | GEARS | 10.7 (12.1) | 3.5 (2.0) | 14.0 (14.0) | 20.7 (24.0) |
| | RECOVER | **4.7 (0.5)** | **1.9 (0.4)** | 5.6 (1.0) | 3.0 (0.4) |
| | NAIAD | **4.7 (0.1)** | **1.9 (0.2)** | **5.4 (0.6)** | **2.8 (0.6)** |

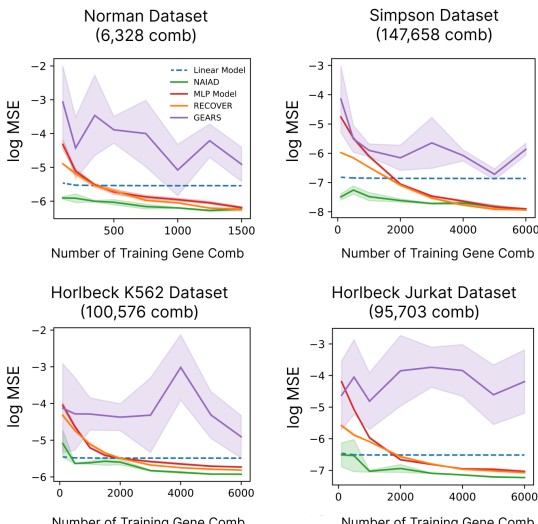

*Figure 3.* Benchmark analysis comparing the NAIAD model with GEARS and RECOVER models, evaluated using test data log(MSE) across different numbers of gene combinations in training data. Error bars are SE across three cross-fold replicates.

son correlation coefficient, and true positive rate from the held-out test data (Figure 3, Appendix Figure 7).

Due to variations in the total number of gene pairs measured across these datasets, we also analyzed model performance based on the average frequency of each gene's occurrence among the training data combinations. Gene occurrence was approximated as Gene Occurrence $= \frac{2N}{M}$ , where $N$ is the number of gene combinations in the training set (the factor of 2 assumes a symmetrical screen), and $M$ is the number of unique genes covered in the screen. We observed that when each gene was seen on average four times, NA-

IAD's performance was consistently the best—over 40% better than the second-best model on average across the four datasets (Table 1) based on root mean square error (RMSE). As the frequency of gene occurrence increased, the performance difference among the models gradually decreased (Table 1, Appendix Table 6). When each gene appeared 20 times within different gene combinations in the training dataset, all models achieved comparable performance levels (Table 1). This suggests that as more data becomes available, the gene embeddings learned from different deep learning models can all largely capture the genetic interactions that dominate performance. Notably, because Gene Occurrence depends on the number of unique genes included in the screen, achieving a high frequency of observations per gene becomes increasingly challenging when the aim is to cover a wide range of genes. For example, screening 20,000 genes across the genome to obtain an average of 20 observations per gene would require measuring 200,000 combinations in the initial training dataset. This represents a substantial experimental cost and even exceeds the size of the largest current screen, which covers over 145,000 combinations (Simpson et al., 2023).

These results demonstrate the robustness of NAIAD in handling both small and large training datasets, making it a valuable tool for exploring vast combinatorial spaces. This highlights its potential for discovery of effective gene combinations in settings with constrained experimental resources.

### 5.3. Effectiveness of Maximum Predicted Effect Sampling in Identifying Effective Gene Pairs

The second challenge in active learning frameworks is the design of the recommendation system or acquisition function. We explored multiple acquisition functions—including uncertainty sampling, Maximum Predicted Effect (MPE)

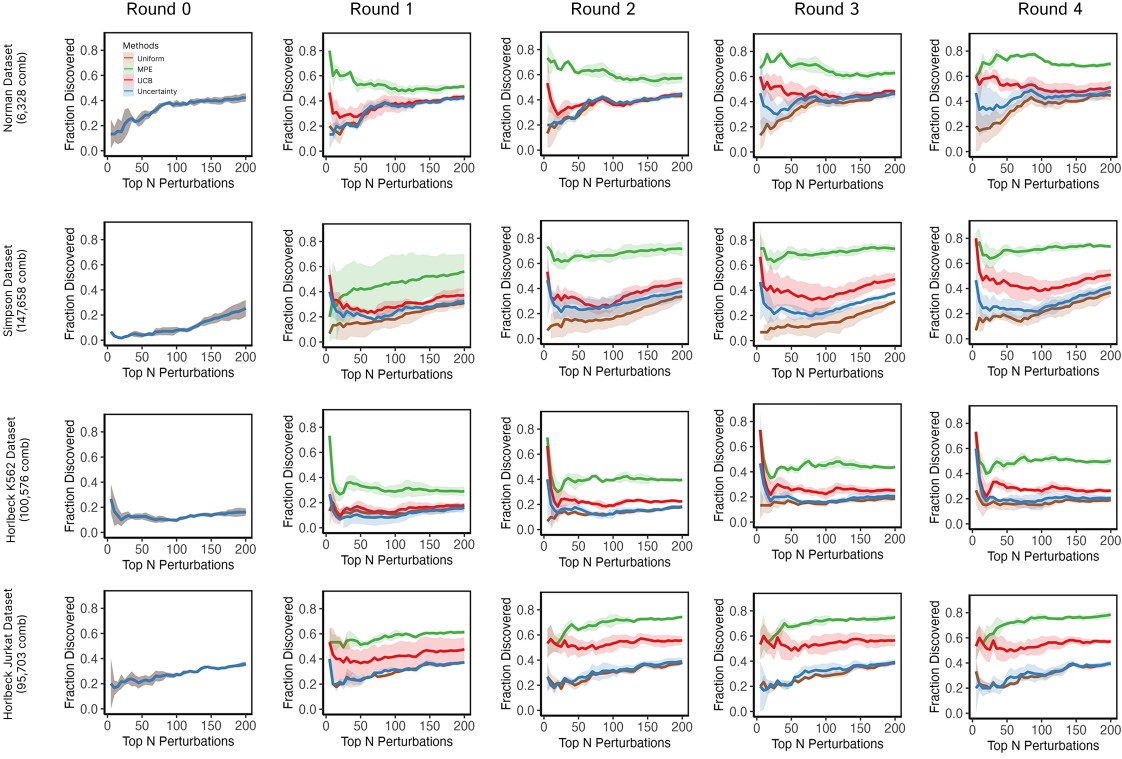

*Figure 4.* Comparison of different acquisition functions evaluated by top-$N$ prediction accuracy for the strongest $N$ perturbations across four iteration rounds (see Appendix D for full description of accuracy metric). Error bars are SE from three cross-fold replicates.

*Table 2.* Number of top 200 gene combinations correctly discovered at round 4 of active learning.

|             | Norman | Simpson | Horlbeck K562 | Horlbeck Jurkat |
|-------------|--------|---------|---------------|-----------------|
| Uniform     | 92     | 66      | 39            | 79              |
| MPE         | **144** | **144** | **94**        | **161**         |
| UCB         | 97     | 111     | 55            | 117             |
| Uncertainty | 92     | 74      | 45            | 84              |

*Table 3.* Marginal gain of correctly discovered top 200 gene combinations at round 4 of active learning.

|             | Norman | Simpson | Horlbeck K562 | Horlbeck Jurkat |
|-------------|--------|---------|---------------|-----------------|
| Uniform     | 1      | 7.5     | 0.5           | 2.5             |
| MPE         | **14** | **27**  | **14.25**     | **23**          |
| UCB         | 2.25   | 18.75   | 4.5           | 12              |
| Uncertainty | 1      | 9.5     | 2             | 3.75            |

sampling, and Upper Confidence Bound (UCB) sampling from RECOVER (Bertin et al., 2023), which combines residual and uncertainty sampling—to evaluate their impact on overall performance (Figure 4). To simulate the active learning process using current publicly available symmetrical screening data, we started with the same uniformly sampled gene pairs for all acquisition functions and trained a baseline ensemble of NAIAD models. We then used the trained model ensemble to infer unseen combinations across the entire combinatorial space and applied different acquisition functions on the corresponding ensemble metric (e.g. MPE or ensemble uncertainty) to select gene pairs for measurement as the additional training data for the next round. We repeated the sampling and retraining process across four iterations.

We found that MPE sampling outperforms other sampling methods by identifying a higher fraction of the globally strongest gene pairs (Figure 4). The advantage of MPE sampling becomes larger with each additional iteration, even though all approaches show improved performance over iterations. By the fourth iteration, across the four datasets, we were able to uncover over twice as many strong perturbations using MPE sampling compared to uniform sampling, and nearly 1.5 times as many as UCB, the second-best method (Table 2). Specifically, the MPE method identified approximately 150 out of the top 200 strongest gene pairs in three of the datasets, achieving the highest marginal gain in each dataset (Table 3). We found this result to be robust under different active learning ensemble sizes, ranging from one model per ensemble up to seven models per ensemble (Appendix Table 5).

*Table 4.* Number of top 200 gene combinations correctly discovered at round 4 of active learning using NAIAD and RECOVER with different acquisition functions. Error is SE across three cross-fold replicates.

| Model | Method | Norman | Simpson | Horlbeck K562 | Horlbeck Jurkat |
|---|---|---|---|---|---|
| NAIAD | Uniform | 93.3 (0.7) | 70.3 (5.0) | 38.7 (0.3) | 81.3 (0.9) |
| RECOVER | Uniform | 81.0 (0.0) | 44.3 (0.3) | 37.0 (0.6) | 75.0 (0.6) |
| NAIAD | MPE | **143.0 (1.5)** | **141.7 (3.7)** | **99.7 (4.2)** | **150.0 (4.0)** |
| RECOVER | MPE | 138.7 (2.7) | 88.3 (28.2) | 53.7 (23.7) | 65.7 (10.4) |
| NAIAD | UCB | 110.3 (2.5) | 102.7 (8.1) | 60.3 (3.3) | 96.0 (8.5) |
| RECOVER | UCB | 84.7 (3.5) | 62.0 (14.1) | 26.3 (1.3) | 48.7 (7.1) |

Although MPE sampling exhibited worse overall performance based on the test set MSE (Appendix Figure 8), this discrepancy is likely due to the distribution of the sampled training data. MPE sampling skews the distribution toward strong-effect gene pairs, leading to more accurate predictions for these pairs but less accuracy across the entire dataset, thus affecting overall MSE (Appendix Figure 9).

We also considered alternative models and acquisition functions to assess their ability to identify strong pairs of perturbations. Additional experiments performed in an active learning setting using the MPE acquisition function showed that NAIAD outperformed both RECOVER and GEARS (Appendix Table 7). We next examined the effect of combining exploration and exploitation during sampling by incorporating an uncertainty-based score term to the MPE sampling function (see Appendix C.5), inspired by the UCB strategy from RECOVER. This MPE+uncertainty function differs from UCB, which adds an ensemble uncertainty term to the *residual* (subtracting ensemble prediction mean from linear model baseline) rather than the *mean* of the ensemble estimate. We found that including an uncertainty term did not improve performance, and when assigned a high weight in the sampling function, reduced the number of strong perturbations discovered during acquisition (Appendix Table 8).

Furthermore, we disentangle the contributions of the MPE acquisition function and the NAIAD model architecture toward identifying optimal gene pairs (Table 4). We found that the MPE acquisition function robustly identifies strong combinations when using the RECOVER model, and additionally that NAIAD outperforms RECOVER head-to-head in an active learning context for every acquisition strategy evaluated. The NAIAD model and MPE acquisition function Individually increased our ability to find effective combinations, and they jointly identified more top-performing gene pairs than any other model-function pairing. Across the four gene perturbation datasets, after four rounds of active learning, NAIAD+MPE outperformed RECOVER+MPE by 1.7 times on average, and NAIAD+UCB by 1.5 times on average.

The primary goal of our NAIAD active learning framework is not to accurately predict all gene pair interactions but to effectively select the strongest gene pairs that induce significant phenotypic changes. Therefore, the superior performance of NAIAD+MPE sampling in identifying potent gene combinations aligns well with our objectives. By using a model that most effectively represents genetic perturbation interactions, and prioritizing the discovery of the strongest gene pairs, we can accelerate the identification of gene combinations that are most relevant for therapeutic development.

## 5.4. Extending NAIAD for Drug Combination Predictions

To assess NAIAD's effectiveness beyond genetic interactions, we applied our framework to drug combination predictions. Drug combination screening is a commonly used approach for identifying optimal combination therapies that can enhance treatment efficacy while minimizing toxicity. We extended NAIAD to drug combination using the publicly available Zheng et al. (2021) dataset, which was adapted by the Bertin et al. (2023) RECOVER study (details in Appendix A). Starting with randomized drug embeddings for the small molecules included in the screen, we incorporated MPE as the acquisition strategy.

We benchmarked NAIAD against RECOVER in an active learning framework for drug combination screening, evaluating performance improvements over successive experimental iterations. As the number of experimental iterations increased, all methods showed performance gains. NAIAD consistently outperformed RECOVER in identifying the strongest drug combinations, and the performance gap increased significantly starting from Round 2. This suggests that NAIAD learns meaningful drug interaction representations more efficiently than RECOVER, making it particularly advantageous when the number of experimental iterations is limited. In later rounds, we observed comparable performance between the two models when considering the top 50 pairs, but NAIAD more effectively identifies the strongest combinations when considering both fewer than 50 and greater than 50 top combinations. Overall, NAIAD

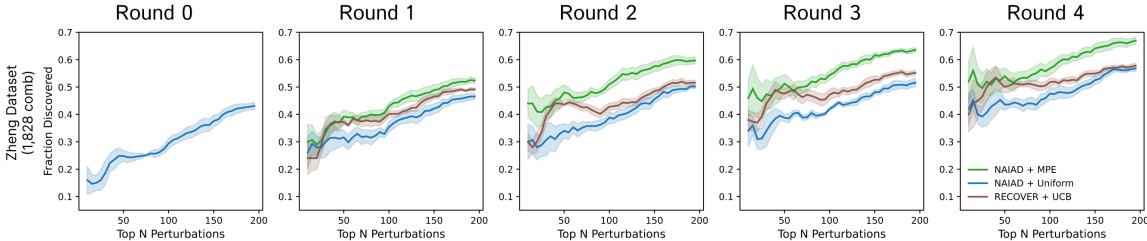

*Figure 5.* Comparison of NAIAD and RECOVER in drug combination screening, assessed by top-$N$ prediction accuracy for the strongest $N$ perturbations across four iteration rounds. The NAIAD model with uniform sampling is used as the baseline model. Error bars are SE from three cross-fold replicates.

is superior to RECOVER in identifying a broader range of effective drug pairs (Figure 5).

These results demonstrate NAIAD's robustness across both gene and drug perturbation domains. Its generalizability, combined with its efficiency in low-data regimes, makes NAIAD a practical tool for discovering effective therapeutic combinations under experimental constraints.

## 6. Discussion and Conclusion

In this work, we introduced NAIAD, an active learning framework that efficiently identifies effective gene or drug pairs by leveraging single perturbation effects, adaptive gene embeddings, and an MPE acquisition function. Our framework leverages the principles of Bayesian optimization (Frazier, 2018), employing sequential experimentation and learning to enable effective decision-making with limited data. Whereas traditional Bayesian optimization employs a fixed surrogate model, our framework adapts to learn different surrogate functions at varying sizes of training data. This adaptability enables NAIAD to adjust its modeling complexity based on the available data, effectively bridging the gap between linear models and deep learning techniques.

By incorporating adaptive gene embeddings, NAIAD mitigates overfitting in small-sample training and captures complex interactions as more data becomes available. The MPE acquisition function further enhances the model's efficiency by prioritizing gene pairs with significant predicted effects, accelerating the discovery process with fewer experimental iterations. As a result, NAIAD outperforms existing approaches, particularly in scenarios with limited training data when using the MPE acquisition function. We demonstrate its effectiveness in small-sample learning and its high discovery rate for identifying the top $N$ strongest combinations across the entire search space.

Despite its advantages, there are areas where the NAIAD model could be further enhanced. Currently, NAIAD does not support predicting gene expression profiles resulting

from single-cell combinatorial perturbation data. However, when projecting single-cell expression data into one relevant phenotype value, the NAIAD framework can still be adapted to predict that specific phenotype. Another limitation is that our model assumes we have knowledge of each gene's individual effect on the phenotype. For unseen individual genes, the current NAIAD model cannot predict the effects of combinations involving these genes, especially when both genes in a pair are unseen. Incorporating properly pre-trained gene embeddings could potentially allow us to predict such unseen situations (Cui et al., 2024). By incorporating pre-trained gene embeddings, we can obtain prior knowledge of the similarities between unseen and known genes within relevant latent spaces. Assuming that these gene relationships are conserved across different domains, the model can leverage this information to infer the single-gene effects of unseen genes, even to predict the outcomes of combinations involving two previously unseen genes. Additionally, refining the acquisition score function to be a learnable component—such as employing a monotonic submodular regularization (Alieva et al., 2020; Wei et al., 2015; Golovin & Krause, 2010)—could enable the model to adaptively prioritize experiments that maximize information gain, rather than relying on a fixed heuristic.

Our model also holds significant potential for higher-order combinatorial perturbations. With the advancement of combinatorial CRISPR technologies, higher-order gene combination datasets are becoming increasingly common (Tieu et al., 2024; Hsiung et al., 2024). NAIAD is theoretically well-suited and can be easily adapted to accommodate interactions beyond gene pairs. Handling higher-order CRISPR combinatorial perturbations would allow for the exploration of more complex genetic interactions, potentially leading to a more effective induction of desired cellular phenotypes.

## Software and Data

The source code for NAIAD is available at https://github.com/NeptuneBio/NAIAD.

## Acknowledgements

We thank our colleagues at Neptune Bio for creating a collaborative and supportive environment that enabled us to complete this work.

## Impact Statement

This paper aims to advance the field of Machine Learning with significant implications for biomedical research. NAIAD's active learning framework has the potential to accelerate the discovery of effective combination therapies by identifying strong combinatorial and synergistic targets with enhanced efficacy and reduced toxicity. By optimizing experimental resource allocation, our framework could contribute to more efficient drug development and improved therapeutic strategies.

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

# A. Dataset Summary

## A.1. Genetic Perturbations

We utilize four bulk combinatorial perturbation datasets in our study:

Combinatorial CRISPRa on K562 cells (Norman et al., 2019): This dataset was generated using combinatorial CRISPR activation (CRISPRa), involving 112 genes and 6,328 unique gene combinations.

Large-scale combinatorial CRISPRi on K562 cells (Simpson et al., 2023): This dataset was generated using CRISPR interference (CRISPRi), involving 543 genes and 147,658 unique gene combinations.

Combinatorial CRISPRi on K562 cells (Horlbeck et al., 2018): This dataset was generated using combinatorial CRISPRi, involving 448 genes and 100,576 unique gene combinations.

Combinatorial CRISPRi on Jurkat T Cells (Horlbeck et al., 2018): This dataset was generated using combinatorial CRISPRi, involving 437 genes and 95,703 unique gene combinations.

For the bulk cell viability datasets, we calculated the log-fold change in cell viability for each CRISPR-guide combination compared to negative control treatments. This normalization allows for the quantification of the effect size of each perturbation on cell survival. Next, we averaged these cell viability measurements across all pairs of guides targeting each unique pair of genes, to generate gene-level measurements of cell viability. To calculate the effect of single-gene perturbations, we used all guide pairs consisting of gene-targeting guide and one non-targeting guide. For a deeper explanation of calculating cell viability (also referred to as $\gamma$), we refer the reader to Simpson et al. (2023).

## A.2. Drug Treatment

We utilize the drug dataset prepared by Zheng et al. (2021) and used by Bertin et al. (2023) in RECOVER to benchmark their active learning platform. The dataset was generated by assaying 2,320 cell lines with single and two-drug treatments. For benchmarking NAIAD, we use the 4,349 unique two-drug treatments conducted on K562 cells.

We processed the data in several ways:

1. The drug database contains different dosage treatments for each drug on the K562 cell line. Since drugs may have dosage-dependent treatment effects, we filter the assays to only include a single dosage for each drug. For each drug, we choose the dosage closest to the IC50 of that drug in K562 cells, and discard all measurements at other dosages.

2. We only use combinations for which the corresponding single-drug measurements (at the same dosages) are also present in the dataset.

3. We remove conditions that saturate the cell viability measurements, i.e. combinations and individual drugs with > 80% cell viability inhibition. We rationalize that at this detection limit of the experiment, a) the measurements are overly-noisy as a result of low cell counts, and b) there are no meaningful interactions between drugs since all cells are dead under each individual treatment.

After these filtering steps, we are left with 1,828 unique measurements in K562 cells across combinations of 94 unique drugs.

# B. Benchmark Models

**Linear model**: We employed a simple linear regression model using the effects of two single-gene perturbations as independent variables to predict their joint effect. This model involved three parameters: two coefficients for the individual gene effects and an intercept term.

**Multi-layer perceptron (MLP)**: We implemented an MLP model where each gene was represented by a 128-dimensional embedding. For each two-gene perturbation, we created a joint embedding for the two genes by summing their embeddings along each dimension and passing it through a MLP layer that projects from 128 dimensions down to a single dimension corresponding to the phenotype value. The MLP captured non-linear interactions between gene embeddings.

**GEARS**: For the GEARS model, we adhered to the original settings as specified in their supplementary materials (Roohani

et al., 2023) for cell viability prediction. This included their specific network architecture and parameters used in their tutorial. For bulk-combinatorial datasets lacking single-cell experiments expected for GEARS, we generated synthetic Pertub-seq datasets of normalized gene expression matrices, with a separate Gaussian $\mathcal{N}(0,1)$ used for sampling the expression of each gene.

**RECOVER adaptation**: RECOVER was originally developed for drug perturbations involving small molecule embeddings and bilinear operations to combine drugs (Bertin et al., 2023). The models were trained to predict Bliss synergy scores, which capture the non-linear components of phenotypic outcomes, instead of directly modeling the phenotypic outcomes themselves. We adapted RECOVER for gene perturbations by incorporating a bilinear projection module to combine 128-dimensional gene embeddings. This adaptation allowed us to model gene-gene interactions using the same principles applied to drug combinations in the original RECOVER framework. We train the model to predict the gene-equivalent of Bliss scores, which is the difference between measured viability and the product of the viability of single-gene perturbations. We then convert the gene-Bliss-score predictions back to overall cell viability predictions by re-adding the product of the viability of the single-gene perturbations.

## C. Active Learning Sampling Strategies

In our active learning framework, we employ several sampling strategies to select the most informative experiments for subsequent rounds. Below, we provide the mathematical formulations for these strategies.

### C.1. Uncertainty-Based Sampling

We use an ensemble of models $\{M_i\}_{i=1}^N$, each initialized randomly. For a candidate gene combination $x$, each model provides a prediction $y_i = M_i(x)$. We estimate the prediction uncertainty using the standard deviation of the ensemble predictions:

$$\text{SD}(y) = \sqrt{\frac{1}{N} \sum_{i=1}^N (y_i - \bar{y})^2}$$

where $\bar{y}$ is the mean prediction across the ensemble:

$$\bar{y} = \frac{1}{N} \sum_{i=1}^N y_i.$$

The acquisition function for uncertainty-based sampling is defined as:

$$a_{\text{uncertainty}}(x) = \text{SD}(y).$$

We select the candidate combinations with the highest $a_{\text{uncertainty}}(x)$ values, as they represent samples where the model is most uncertain and additional data could significantly improve the model.

### C.2. Maximum Predicted Effect (MPE) Sampling

This strategy targets gene combinations predicted to have strong effects. The acquisition function is:

$$a_{\text{MPE}}(x) = \bar{y},$$

where $\bar{y}$ is the mean prediction as defined above. We select candidates with the highest absolute predicted effects $\bar{y}$, prioritizing experiments likely to yield substantial biological insights or therapeutic benefits.

### C.3. Residual-Based Sampling

To identify areas where nonlinear interactions are significant and the model may not have fully captured them, we compute the residual between predictions from a non-linear model and a linear model. Let $N(x)$ be the prediction from the non-linear

model and $L(x)$ be the prediction from a linear approximation. The residual is calculated as:

$$r(x) = |N(x) - L(x)|.$$

The acquisition function for residual-based sampling is:

$$a_{\text{residual}}(x) = r(x).$$

We select candidate combinations with the highest residuals $a_{\text{residual}}(x)$, focusing on samples where the non-linear effects are most pronounced and the model's predictions differ significantly from linear expectations.

### C.4. Upper Confidence Bound (UCB) Sampling

We perform UCB sampling following the strategy provided by RECOVER (Bertin et al., 2023):

$$a_{\text{UCB}}(x) = a_{\text{residual}}(x) + \kappa * a_{\text{uncertainty}}(x)$$

Where $\kappa$ is a hyperparameter set to $\kappa = 1$ based on the recommendation from the RECOVER paper.

By combining these sampling strategies, we aim to efficiently explore the combinatorial space of gene interactions. The goal is to maximize information gain with each experimental round, ultimately using a minimal number of experiments to identify effective gene combinations that can induce desired cellular phenotypes.

### C.5. Mean Upper Confidence Bound (Mean-UCB) Sampling

Inspired by the UCB acquisition function from RECOVER, we develop an exploitation-exploration balanced acquisition function that utilizes the mean of the ensemble prediction rather than the residual relative to a linear model. We call this sampling method Mean Upper Confidence Bound (Mean-UCB) sampling. We express the acquisition function as a weighted combination:

$$\text{score} = \alpha \times \bar{y} + \beta \times \text{SD}(y)$$

where $\bar{y}$ is the mean across the ensemble predictions, and $\text{SD}(y)$ is the standard deviation across ensemble predictions.

This unified view captures multiple acquisition strategies:

- **Uniform sampling:** $\alpha = \beta = 0$

- **Uncertainty-only (pure exploration):** $\alpha = 0$

- **MPE (pure exploitation):** $\beta = 0$

- **UCB (balanced trade-off):** $\alpha = \beta = 1$

We evaluate the performance of the acquisition function on identifying the strongest perturbations in Appendix Table 8.

## D. Model Evaluation Metrics

We use several metrics to evaluate the performance of our models.

**MSE**: Mean Square Error

$$\frac{1}{N_{\text{pairs}}} \sum_{i,j \in \text{genes}} (y_{i+j}^{\text{pred}} - y_{i+j}^{\text{truth}})^2$$

**Fraction Discovered**: Through active learning, we want to identify how many of the top $P$ strong perturbations have been identified by our model of interest. Let $M$ be the total number of rounds of active learning, and let $\mathbf{N} = [n_0, n_1, \ldots, n_M]$ be the list that defines how many samples $n_i$ are used for each active learning round $i = 0, 1, \ldots, M$.

In each round of learning, we train on the $n_i$ unmasked samples chosen depending on the acquisition function. After training, we make predictions for the entire unseen dataset. We concatenate the predictions on the unseen data with the unmasked measured values from the seen data, and call this combined set of predictions and measurements $X_i$ for each round $i$.

Let $X_{i,P}$ be the set of top $P$ values in the set $X_i$. Let $Y_{i,P}$ be the set of top $P$ measured (ground truth) values across all samples in round $i$. Let $P_i = |X_{i,P} \cap Y_{i,P}|$ be the number of times a perturbation from the top $P$ predictions is also in the top $P$ targets for round $i$. Fraction Discovered is then defined as $\frac{P_i}{P}$.

**Marginal Gain**: The marginal gain at round 4 is $\frac{P_4(top=200) - P_0(top=200)}{4}$ , where $P_4(top = 200)$ is the number of top-200 perturbations correctly identified in Round 4, and $P_0(top = 200)$ is the number of top-200 perturbations correctly identified in Round 0.

**TPR**: True positive rate

Let $X^{\text{targets}}$ be the set of top $N$ measured combinations, and $X^{\text{preds}}$ be the set of top $N$ predicted combinations. We find the number of matches $N_{\text{match}} = |X^{\text{targets}} \cap X^{\text{preds}}|$, and calculate TPR as $\frac{N_{\text{match}}}{N}$.

# E. Hardware Configuration

All model training was done on a single Paperspace A100-80G server with 100GB of RAM.

# F. Hyperparameter Selection

For all training of the NAIAD and MLP models, we use learning_rate $= 10^{-2}$, and a batch_size $= 1024$. We also used a linear rate scheduler with 10% of training steps used for warm up, and weight_decay $= 0$.

To identify these optimal hyperparameters, we testing hyperparameters across the following ranges:

n_epoch: $[50, 100, 200, 500, 1000, 2000]$

batch_size: $[512, 1024, 2048, 4096]$

learning_rate: $[10^{-4}, 10^{-3}, 10^{-2}, 10^{-1}]$

d_embed: $[2, 4, 8, 16, 32, 64, 128, 256]$

d_single-gene: $[8, 16, 32, 64, 128, 256, 512]$

weight_decay: $[0, 10^{-4}, 10^{-3}]$

For training the RECOVER model, we use a learning_rate $= 10^{-1}$. For GEARS, we keep all hyperparameters assigned by default from the package.

### F.1. Adaptive Model Embedding Size Selection

For the NAIAD model, we choose the embedding size hyperparameter based roughly on the number of times each gene was seen in the training set, following the schedule shown in Table 9 . We determined these values using the idea that model size depends on the number of times each individual gene is seen during training, and that these properties hold across all datasets.

## G. Supplemental Figures and Tables

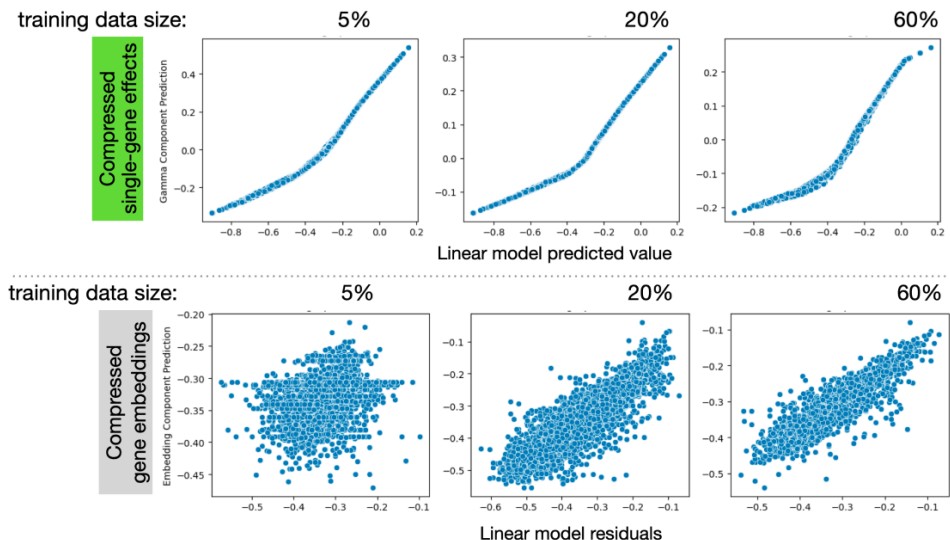

*Figure 6.* Evaluation of compressed single-gene effects and gene embeddings shows a strong correlation between the compressed single-gene effects and the values predicted by the linear model. As the training data increases, the correlation between gene embeddings and the residuals of the linear model predictions gradually becomes stronger.

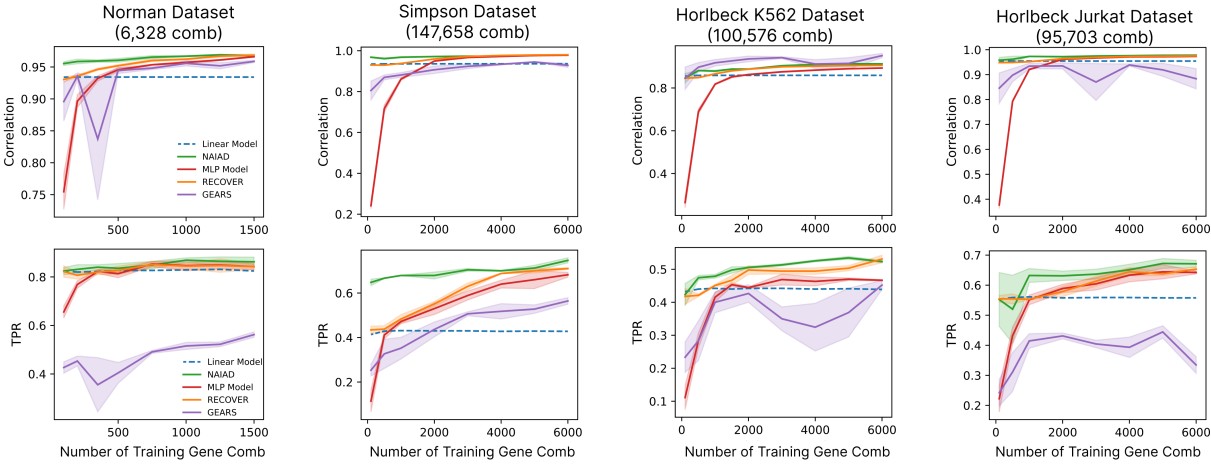

*Figure 7.* Benchmark analysis comparing the NAIAD model with GEARS and RECOVER models, evaluated using test data correlation and true positive rate (for identifying the top 200 perturbations of a 30% held-out test set) across different numbers of gene combinations in training data. Error bars are SE across three cross-fold replicates.

*Table 5.* Number of top 200 gene combinations correctly discovered at round 4 of active learning using different ensemble sizes. Error is standard error (SE) across three cross-fold replicates.

| Ensemble Size | Norman | | Simpson | | Horlbeck K562 | | Horlbeck Jurkat | |
|---|---|---|---|---|---|---|---|---|
| | MPE | Uniform | MPE | Uniform | MPE | Uniform | MPE | Uniform |
| 1 | 140.7 (2.3) | 89.3 (2.0) | 149.7 (2.7) | 66.7 (4.7) | 105.7 (3.9) | 34.3 (5.2) | 153.3 (3.8) | 79.7 (0.7) |
| 3 | 141.3 (1.9) | 91.0 (1.2) | 146.3 (3.8) | 67.7 (4.6) | 104.3 (4.3) | 37.7 (0.9) | 153.3 (5.4) | 79.7 (1.5) |
| 5 | 141.0 (2.5) | 91.0 (1.0) | 143.3 (2.6) | 68.0 (2.6) | 103.7 (3.8) | 37.0 (1.2) | 150.7 (2.8) | 80.7 (1.5) |
| 7 | 142.7 (1.8) | 91.7 (0.7) | 143.3 (3.0) | 71.3 (4.4) | 101.7 (5.8) | 38.7 (0.9) | 151.7 (3.8) | 81.0 (2.5) |

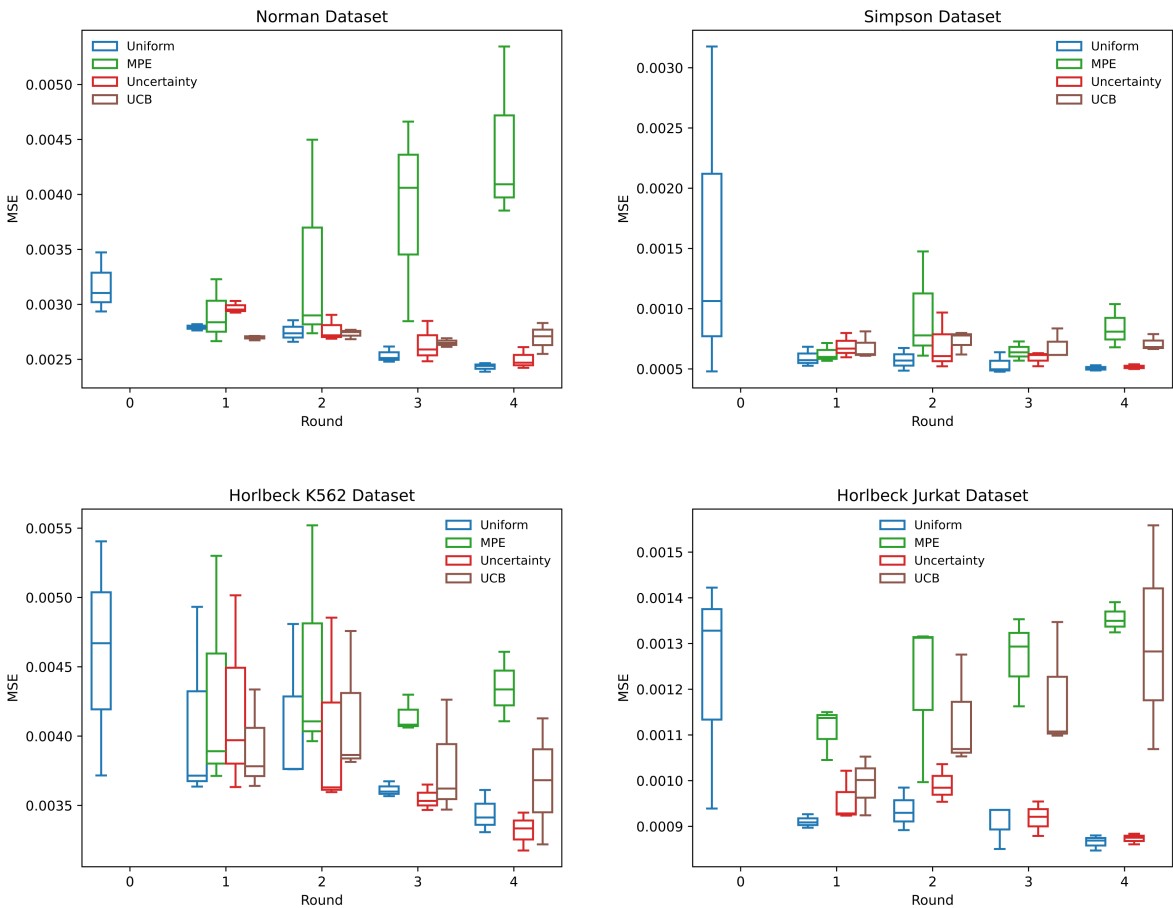

*Figure 8.* Comparison of different acquisition functions evaluated by MSE across four iteration rounds. Error bars are SE across three cross-fold replicates.

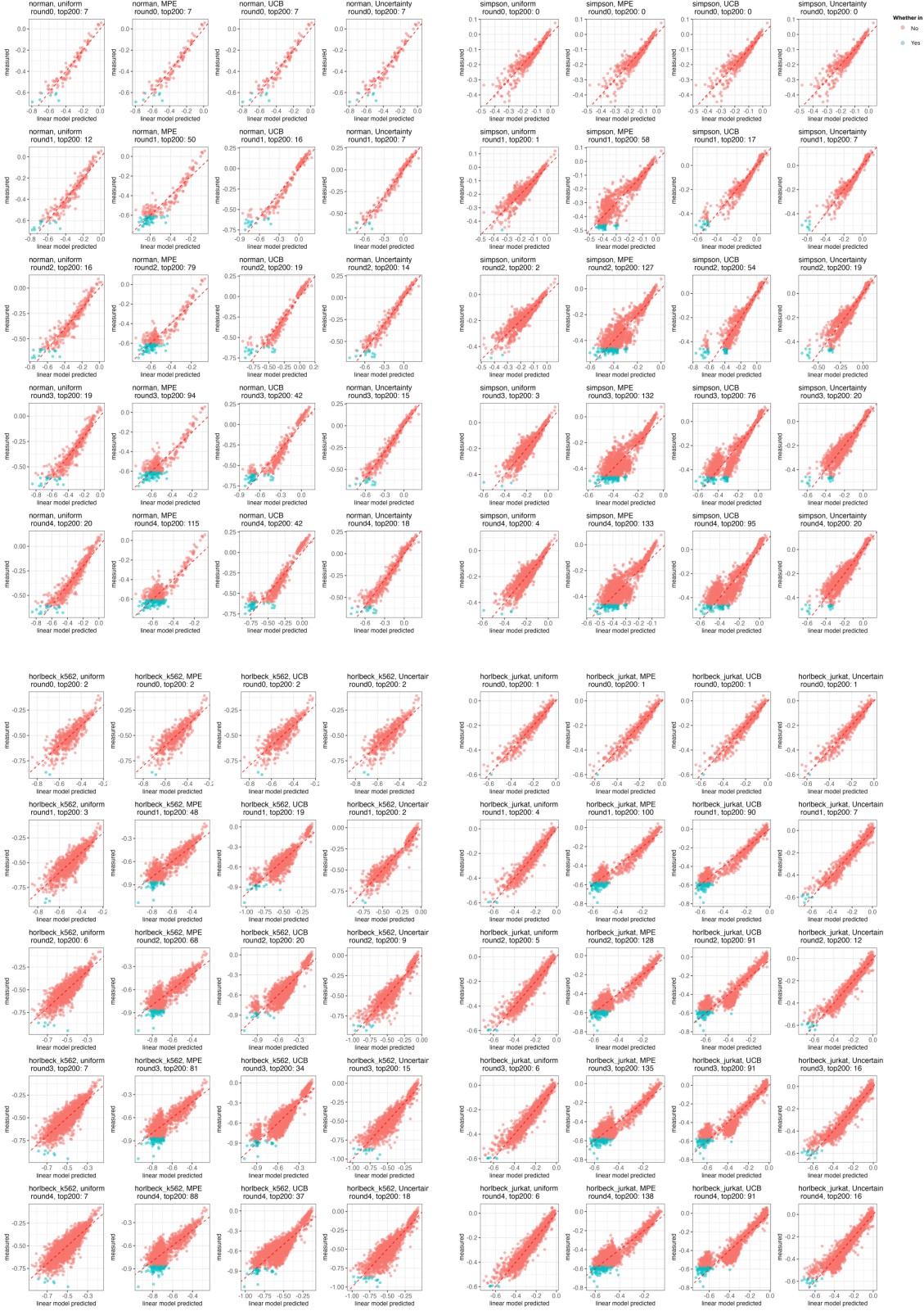

*Figure 9.* Comparison of training sample distribution changes across different iterations using four acquisition functions. The X-axis represents the linear predicted values, while the Y-axis shows the actual measured values. The plot highlights gene pairs belonging to the top 200 gene pairs with the strongest effects in the entire dataset.

*Table 6.* RMSE of five models on test data when each gene is approximately observed in training data 10 times. Error is SE across three cross-fold replicates.

| Gene Frequency | Model | Dataset RMSE ($\times 10^{-2}$) | | | |
|---|---|---|---|---|---|
| | | Norman | Simpson | Horlbeck K562 | Horlbeck Jurkat |
| 10 | Linear | 6.1 (1.3) | 3.3 (0.2) | 6.4 (0.9) | 3.8 (0.6) |
| | MLP | 5.7 (1.3) | 2.4 (0.4) | 6.3 (1.1) | 3.5 (1.0) |
| | GEARS | 15.5 (12.8) | 8.7 (10.3) | 12.0 (9.6) | 24.2 (28.5) |
| | RECOVER | 5.9 (1.0) | 2.5 (0.2) | 6.1 (0.2) | 3.7 (1.0) |
| | **NAIAD** | **4.7 (1.2)** | **2.0 (0.2)** | **5.7 (0.8)** | **3.0 (0.8)** |

*Table 7.* Number of top 200 gene combinations correctly discovered over four rounds of active learning by NAIAD, RECOVER, and GEARS on the Norman dataset. Error is SE across three cross-fold replicates

| Model | Method | Round 0 | Round 1 | Round 2 | Round 3 | Round 4 |
|---|---|---|---|---|---|---|
| NAIAD | MPE | 86.0 (1.0) | 100.3 (0.9) | 112.3 (1.7) | 128.0 (2.5) | 143.0 (1.5) |
| RECOVER | MPE | 82.0 (1.0) | 101.0 (1.2) | 110.7 (2.0) | 125.0 (2.5) | 138.7 (2.7) |
| GEARS | MPE | 42.0 (2.1) | 80.0 (1.5) | 91.7 (0.7) | 99.7 (0.3) | 109.3 (1.5) |
| NAIAD | Uniform | 86.0 (1.0) | 88.3 (2.0) | 88.3 (0.3) | 89.7 (1.2) | 93.7 (0.7) |

*Table 8.* Number of top 200 gene combinations correctly discovered at round 4 of active learning using different $\beta$ values in the Mean-UCB acquisition function. We find that the True Positive Rate (TPR) of the top 200 gene combinations remains relatively stable for small values of $\beta$ (i.e., $\beta \leq 1$). This may be because, during early active learning rounds, the effect magnitude dominates the predictive uncertainty. As $\beta$ increases, the influence of uncertainty grows, leading to a decrease in TPR. Error is standard error (SE) across three cross-fold replicates.

| $\beta$ | Norman | Simpson | Horlbeck K562 | Horlbeck Jurkat |
|---|---|---|---|---|
| 0 | 138.3 (2.2) | 145.7 (3.8) | 99.0 (2.6) | 152.3 (4.1) |
| 0.25 | 139.7 (2.0) | 144.7 (3.7) | 97.0 (3.0) | 151.0 (3.8) |
| 0.5 | 138.3 (0.9) | 148.0 (2.6) | 98.0 (0.6) | 152.0 (4.0) |
| 0.75 | 138.3 (0.9) | 147.3 (2.0) | 98.7 (0.3) | 154.7 (4.9) |
| 1 | 139.3 (1.7) | 144.7 (2.6) | 98.0 (1.0) | 154.3 (4.6) |
| 1.5 | 140.7 (0.3) | 144.3 (2.3) | 97.7 (2.9) | 153.3 (4.6) |
| 2 | 138.7 (2.9) | 144.0 (3.6) | 95.7 (1.9) | 153.7 (4.5) |
| 5 | 137.0 (4.7) | 144.0 (2.6) | 81.3 (1.8) | 143.3 (4.8) |
| 10 | 133.0 (3.6) | 142.3 (2.6) | 72.3 (6.4) | 132.0 (6.9) |
| 25 | 119.7 (3.4) | 124.0 (4.5) | 73.0 (5.8) | 107.7 (1.8) |
| Uniform | 93.3 (2.9) | 69.0 (1.1) | 39.0 (0.6) | 80.3 (0.3) |

*Table 9.* Training and model hyperparameters for each data set and data split

| Dataset | Train Epochs | Avg. Times All Genes Seen | Approx. Training Dataset Size | Embed Dim | Single-Gene Dim |
|---|---|---|---|---|---|
| Norman et al. (2019) | 500 | 2 | 100 | 2 | 64 |
| | | 4 | 200 | 4 | 64 |
| | | 10 | 500 | 16 | 64 |
| | | 20 | 1000 | 16 | 64 |
| | | 30 | 1500 | 32 | 64 |
| | | 40 | 2000 | 64 | 64 |
| | | 60 | 3000 | 64 | 64 |
| | | 80 | 4000 | 128 | 64 |
| | | 100+ | 5000+ | 128 | 64 |
| Simpson et al. (2023) | 200 | 2 | 500 | 2 | 256 |
| | | 4 | 1000 | 4 | 256 |
| | | 10 | 5000 | 16 | 256 |
| | | 20 | 10000 | 16 | 256 |
| | | 30 | 15000 | 32 | 256 |
| | | 40 | 20000 | 64 | 256 |
| | | 60 | 30000 | 64 | 256 |
| | | 80 | 40000 | 128 | 256 |
| | | 100+ | 50000+ | 128 | 256 |
| Horlbeck et al. (2018) | 200 | 2 | 500 | 4 | 256 |
| | | 4 | 1000 | 16 | 256 |
| | | 10 | 5000 | 32 | 256 |
| | | 20 | 10000 | 32 | 256 |
| | | 30 | 15000 | 64 | 256 |
| | | 40 | 20000 | 64 | 256 |
| | | 60 | 30000 | 64 | 256 |
| | | 80 | 40000 | 128 | 256 |
| | | 100+ | 50000+ | 128 | 256 |
| Zheng et al. (2021) | 800 | 2 | 100 | 2 | 256 |
| | | 4 | 200 | 4 | 256 |
| | | 6 | 300 | 4 | 256 |
| | | 8 | 400 | 4 | 256 |
| | | 10 | 500 | 16 | 256 |

