# OpenReview forum: "Active Learning for Efficient Discovery of Optimal Combinatorial Perturbations"
_ICML.cc/2025/Conference — ICML 2025 poster_

### Official Review · Reviewer_PDeb · 2025-03-12

**Overall Recommendation:** 3

**Summary:**

This work proposes a new active learning framework for optimizing desirable properties in CRISPR screening, which is a combinatorial problem.
The main algorithmic contribution comes down to an adaptive training method that scales the embedding dimensionality of the predictive model with the size of the training size, which allows the representation to learn the necessary characteristics of the genes as the active learning loop progresses.
The authors conducted experiments to compare the performance of the proposed algorithm against various baselines.

## update after rebuttal

The authors have sufficiently responded to my questions, and I will increase my score.

**Claims And Evidence:**

Yes, the claims are appropriately backed by the experiments.

**Essential References Not Discussed:**

N/A

**Experimental Designs Or Analyses:**

Yes, the experimental designs are appropriate.
The analyses could be extended to get a better insight into what makes the active learning policy works as well as the scalability of the method.
For example, as I understand, MPE is a version of UCB that does not explore (setting the tradeoff parameter $\beta = 0$, simply optimizing the predicted mean).
If so, I find it interesting that this version outperforms UCB with a non-zero $\beta$.
Do the authors have any comments on whether other values for $\beta$ have been tried and what the trend is.

In terms of scaling the dimensionality of the embedding with the training data size, do we do this at every step of the active learning loop?
This seems to be quite an expensive endeavor, as we need to train multiple deep neural networks (since an ensemble is used) every time new data comes in.
On this note, have the authors considered how performance changes as a function of the size of the ensemble?

Finally, could the authors provide some discussion regarding how scalable for the number of perturbations to be larger than 2? Does it take exponentially more effort to extend this framework?

**Methods And Evaluation Criteria:**

Yes, the proposed methods and evaluation make sense.

**Other Comments Or Suggestions:**

Is there a reason why the different curves start off at different points in the plots?

**Other Strengths And Weaknesses:**

N/A

**Questions For Authors:**

N/A

**Relation To Broader Scientific Literature:**

The proposed method is a new active learning framework for optimizing over gene combinations which outperforms baselines from previous works.

**Theoretical Claims:**

There are no theoretical claims.

---

> ### Author Rebuttal · Authors · 2025-04-01
>
> Dear Reviewer PDeb,
>
> We greatly appreciate your thoughtful feedback and questions. Your comments have helped us identify key areas where we could improve the presentation and clarity of our work. We will incorporate the additional results from the experiments in the revised manuscript. Below, we address your questions point by point. Due to space constraints, we have the full tables with standard errors here: https://bit.ly/42bZt32.
>
> ---
> > [Q1] Do the authors have any comments on whether other values for β have been tried and what the trend is?
>
> [A1] You raised an important point regarding the role of β in the acquisition function. This prompted us to explore a more generalized formulation of our acquisition function. We now express the acquisition score as a weighted combination:
>  score = α × mean + β × std.
> This unified view captures multiple strategies:
>
> Uniform sampling: α = β = 0
>
> Uncertainty-only (pure exploration): α = 0
>
> MPE (pure exploitation): β = 0
>
> UCB: α = β = 1 (equal trade-off)
>
> To better understand how this trade-off impacts performance, we added a new experiment that varies β while holding α = 1, testing β values from 0 to 25 on four genetic perturbation datasets.
>
> We find that the True Positive Rate (TPR) of top 200 gene combinations remains relatively stable for small β (β ≤ 1). This may result from, in early rounds, the effect magnitude dominating the predictive uncertainty. As β increases, uncertainty begins to dominate and leads to a decrease in TPR.
> |β|Norman|Simpson|Horlbeck K562|Horlbeck Jurkat|
> |-|-|-|-|-|
> |0|138|146|99|152|
> |0.5|138|148|98|152|
> |1|139|145|98|154|
> |5|137|144|81|143|
> |10|133|142|72|132|
> |25|120|124|73|108|
> |Uniform|93|69|39|80|
>
> We also want to clarify that UCB in RECOVER is defined differently: it uses residuals + β × std, since the model’s objective is to predict residuals from a linear baseline (synergy prediction). We will revise the manuscript to distinguish these UCB formulations more clearly.
> Finally, we expanded our experiments to compare the effectiveness of MPE and residual-UCB strategies across both NAIAD and RECOVER. The results are in our response to Reviewer kwgr [see A4]. Consistently, MPE outperforms residual-based UCB methods, further supporting our conclusion.
>
> > [Q2] Is embedding dimensionality scaled at every active learning step? Given the expensive cost of retraining multiple networks, have the authors evaluated performance versus ensemble size?
>
> [A2]  You're correct that we increase the embedding size during the active learning iterations. Retraining the ensemble from scratch each round is expensive (typically hours). However, this remains fast compared to the time and cost of biological experiments (weeks per cycle).
>
> Your suggestion to test the impact of ensemble size was valuable. We performed an experiment varying the number of ensemble models. The results indicate that increasing the ensemble size can improve performance under uniform sampling. In contrast, the MPE acquisition function demonstrates robust performance regardless of ensemble size.
> |Ensemble Size|Norman||Simpson||Horlbeck K562||Horlbeck Jurkat||
> |-|-|-|-|-|-|-|-|-|
> ||MPE|Uniform|MPE|Uniform|MPE|Uniform|MPE|Uniform|
> |1|141|89|150|67|106|34|153|80|
> |5|141|91|143|68|104|37|151|81|
> |7|143|92|143|71|102|39|152|81|
>
> > [Q3] How scalable for the number of perturbations to be larger than 2? Does it take exponentially more effort to extend this framework?
>
> [A3] Thank you for highlighting this valuable point. Our framework extends efficiently to fit combinations with more than two gene perturbations. We will revise the manuscript to provide a more detailed explanation.
> For a dataset with a higher-order combination involving p perturbations, we define the set $S=\\{i_1,i_2,…,i_p\\}$. We model the effect of jointly perturbing the genes in $S$ as:
> $$Y_{S} = f_1([Y_{i_1}, Y_{i_2}, … Y_{i_p}]W_1)A^T + \sum_{i \in S} f_2(W_2X^i_{gene})A^T_2$$
> Here, $Y_{i_1}, Y_{i_2}, \dots Y_{i_p} $ represent the individual gene effects, while the interaction term $\sum_{i \in S} f_2(W_2X^i_{gene})A^T_2$ captures the high-order genetic interaction effect. We leverage a permutation-invariant mechanism for combining individual gene embeddings through summation, allowing the model to efficiently generalize across different lengths of perturbation sets.
>
> > [Q4]  Is there a reason why the different curves start off at different points in the plots?
>
> [A4] Thanks for pointing this out. In Figure 4, the x-axis should start at “5,” the minimum TPR for the top 5 gene combos. Since strategies choose different samples after round one, the curves diverge from there. We'll update the figure by removing “0” and marking the start as “5.”
>
> ---
> Thank you for your thoughtful and constructive feedback. It has helped strengthen and refine our work. If you have any further questions, we’d be happy to address them. If you feel all concerns have been resolved, we kindly invite you to consider re-rating your evaluation. Thank you.

---

### Official Review · Reviewer_kwgr · 2025-03-13

**Overall Recommendation:** 4

**Summary:**

We can perturb the expression of various genes to achieve a desirable phenotype such as enhanced cell viability. However, given that there are close to 20k known human genes, it is not possible to test every gene combination to identify the optimal combination that leads to the most desirable phenotype. This paper presents an active learning method called NAIAD for discovering optimal gene pairs using existing perturbation data. NAIAD builds a model to predict the effect of jointly perturbing a pair of genes that not only accounts for additive effects but also for interactions been the genes. This model is coupled with an acquisition strategy, the most effective being Maximum Predicted Effects or MPE, to choose gene pairs over active learning iterations. Crucially, NAIAD also explicitly accounts for the need to efficiently model datasets with different sizes over active learning iterations by progressively increasing the dimensionality of gene embeddings as more data is obtained. In their experiments that use gene perturbation and drug combination data, the authors convincingly demonstrate the following:
- Show that adaptive gene embedding sizes allow NAIAD to learn more effective models across training sets of different sizes when compared to using fixed embedding sizes
- NAIAD outperforms existing baselines in modelling combinatorial gene perturbation datasets, especially when training data is limited.
- In the active learning setting, the MPE acquisition function outperforms all other functions and identifies the most number of gene pairs that produce the greatest changes in the measured phenotype.
- NAIAD + MPE outperforms the baseline RECOVER + UCB method in identifying optimal drug combinations.

## Update after rebuttal
The additional results provided in the rebuttal further highlight the effectiveness of NAIAD and help understand the usefulness of MPE alone. All of my questions have been satisfactorily addressed and I will keep my original score.

**Claims And Evidence:**

The paper is very well-written and claims made throughout the paper are supported by clear and convincing evidence. Additionally, shortcomings and ways to improve on NAIAD are transparently described in the discussion section.

**Essential References Not Discussed:**

To the best of my knowledge, relevant prior work has been adequately discussed.

**Experimental Designs Or Analyses:**

Yes, I checked the soundness of all the experiments presented and have mentioned my suggestions above.

**Methods And Evaluation Criteria:**

The method is well-motivated and the benchmarks used are quite comprehensive and meaningful. I have a few suggestions here to help improve the benchmarking:
- Although the drug combination benchmark shows that NAIAD + MPE outperforms RECOVER + UCB, it would be useful to see active learning results from RECOVER and GEARS in the gene combination benchmarks to more convincingly demonstrate that NAIAD outperforms these methods in the active learning setting.
- Figure 5 seems to indicate that using MPE leads to NAIAD outperforming RECOVER. To isolate the benefits of using MPE vs. using the NAIAD model, having results from RECOVER + MPE would useful.

**Other Comments Or Suggestions:**

N/A

**Other Strengths And Weaknesses:**

N/A

**Questions For Authors:**

1. In an active learning setting, how is it handled when the acquisition strategy queries a gene combination that is not present in the dataset? Do you only consider pairs for which data is available during acquisition?
2. How do RECOVER and GEARS perform on the gene combination-based active learning task?
3. How much improvement comes from using MPE alone? Have the authors tried using MPE along with RECOVER in their benchmarking?
4. In lines 129-130, shouldn't $k$ be the total number of genes for the row vector $X^{i}_{gene}$ to represent the $i$th gene's embedding?

**Relation To Broader Scientific Literature:**

The main contribution of this paper is the NAIAD modelling framework that can accurately model effects of perturbations in a pair of genes from limited training data by incorporating both additive and interaction effects. The use of adaptive gene embedding sizes to account for differences in training data sizes across active learning iterations is novel to the best of my knowledge. RECOVER (Bertin et al. (2023)) is the closest prior work but their model does not explicitly have an additive effects term to the best of my knowledge (they only seem to have an interactive effects term). RECOVER was also only used for identifying drug combinations in the original paper. Although the MPE acquisition function is very simple, its use in this setting also appears to be novel and the authors demonstrate its effectiveness.

**Theoretical Claims:**

The paper does not make theoretical claims.

---

> ### Author Rebuttal · Authors · 2025-04-01
>
> Dear Reviewer kwgr,
>
> We appreciate your supportive feedback and your valuable comments, which have helped us to improve our work further. We will incorporate the results from your suggested experiments in the revised manuscript. We address your questions below. Due to space constraints, we have the full tables with standard errors here: https://bit.ly/42bZt32.
>
> ---
>
> > [Q1] how is it handled when the acquisition strategy queries a gene combination that is not present in the dataset? Do you only consider pairs for which data is available during acquisition?
>
> [A1]  We appreciate your point. The datasets used in our work were originally designed to be comprehensive, ensuring that all pairwise gene perturbation combinations were covered. In our simulated active learning framework, the full dataset includes all possible gene combinations. In contrast, during real active learning, each round selects gene pairs that have not been previously measured. These could then be measured through additional biological experiments. We will revise the manuscript to more clearly explain this.
>
> > [Q2] How do RECOVER and GEARS perform on the gene combination-based active learning task?
>
> [A2] Thank you for suggesting a comparison between RECOVER, GEARS and NAIAD in active learning tasks to enhance the benchmark's comprehensiveness. We've added an experiment evaluating RECOVER, GEARS, and NAIAD using MPE on the Norman dataset. The results below, showing the true positive rate (TPR) of the top 200 hits, demonstrate that NAIAD continues to outperform GEARS and RECOVER in active learning rounds.
>
> The GEARS framework does not have active learning iterations and is not efficient for large combinatorial perturbation. Our active learning framework includes ensembles and multiple replicates, further increasing runtime. Thus, we focused the GEARS benchmark on the Norman dataset. A more comprehensive comparison with RECOVER is provided in [A3] and [A4].
>
> |Model|Method|Round0|Round1|Round2|Round3|Round4|
> |-|-|-|-|-|-|-|
> |NAIAD|MPE|86|100|112|128|143|
> |RECOVER|MPE|82|101|111|125|139|
> |GEARS|MPE|42|80|92|100|109|
> |NAIAD|Uniform|86|88|88|90|94|
> > [Q3] Figure 5 seems to indicate that using MPE leads to NAIAD outperforming RECOVER. To isolate the benefits of using MPE vs. using the NAIAD model, having results from RECOVER + MPE would be useful.
>
> [A3] We appreciate your thoughtful suggestion to disentangle the contributions of the MPE acquisition function and the NAIAD model architecture. In response, we added an experiment using the drug combination dataset, evaluating NAIAD and RECOVER paired with different acquisition strategies: uniform sampling, MPE, and residual-UCB (originally used in RECOVER). The results, summarized in the table showing the TPR of the top 200 hits at round 4, indicate that MPE has a stronger impact on performance than the choice of model architecture. Both RECOVER + MPE and NAIAD + MPE achieve comparable results. However, in the genetic perturbation data, we do see the advantages of both the NAIAD architecture and the MPE acquisition function [comments in A4].
>
> |Model|Method|Drug Combination|
> |-|-|-|
> |NAIAD|MPE|131|
> |RECOVER|MPE|133|
> |NAIAD|residual_UCB|121|
> |RECOVER|residual_UCB|120|
> |NAIAD|Uniform|120|
> |RECOVER|Uniform|117|
>
> > [Q4] How much improvement comes from using MPE alone? Have the authors tried using MPE along with RECOVER in their benchmarking?
>
> [A4]  Thank you for your insightful comments to further explore the role of NAIAD and MPE in active learning. We included an experiment comparing the performance of RECOVER and NAIAD across all genetic perturbation datasets by calculating the TPR of the top 200 hits at round 4. These results highlight the advantages of both the NAIAD architecture and the MPE acquisition function. The discrepancy between the drug combination and genetic perturbation results is likely due to the relatively small size of the drug combination dataset, which includes only 1,800+ combinations, a substantially smaller number than in the genetic perturbation datasets.
>
> |Model|Method|Norman|Simpson|Horlbeck K562|Horlbeck Jurkat|
> |-|-|-|-|-|-|
> |NAIAD|Uniform|93|70|39|81|
> |RECOVER|Uniform|81|44|37|75|
> |NAIAD|MPE|143|142|100|150|
> |RECOVER|MPE|139|88|54|66|
> |NAIAD|res_UCB|110|103|60|96|
> |RECOVER|res_UCB|85|62|26|49|
>
> > [Q5] In lines 129-130, shouldn't k be the total number of genes for the row vector X_gene  to represent the i th gene's embedding?
>
> [A5] Yes.  You are correct that k should be the total number of genes within our dataset. Thank you for noticing this typo. We’ve fixed the statement to read:
>
> "Let $X_{\text{gene}} \in \mathbb{R}^{k \times p} $ be the learnable gene embedding matrix, where $k$ is the number of genes perturbed …."
>
> ---
> Many thanks to Reviewer kwgr for your professional, detailed, and valuable reviews! We have done our best to address each of your concerns and hope our response can resolve them. We will actively join the discussion until the end of the rebuttal period.

---

> > ### Comment · Reviewer_kwgr · 2025-04-04
> >
> > Dear authors,
> >
> > Thank you for the additional analyses, I think these results further highlight the effectiveness of your method and help understand the usefulness of MPE alone. All of my questions have been satisfactorily addressed and I will keep my current score.

---

### Official Review · Reviewer_vQva · 2025-03-20

**Overall Recommendation:** 2

**Summary:**

This paper presents an active learning framework that efficiently discovers optimal gene pairs by leveraging single-gene perturbation effects and adaptive gene embeddings. The experiments show that the proposed method achieve better performance than baseline methods.

**Claims And Evidence:**

Yes.

**Essential References Not Discussed:**

No.

**Experimental Designs Or Analyses:**

Yes.

**Methods And Evaluation Criteria:**

Yes.

**Other Comments Or Suggestions:**

No

**Other Strengths And Weaknesses:**

Strengths
1. the proposed method is straightforward with interesting applications. However, I have to admit that I have shallow understanding on this field. I cannot provide a fair evaluation on the novelty.

Weaknesses
1. Considering that this is a ML conference, I think the author should provide a preliminary section in the main content so the general reader can understand the task.

**Questions For Authors:**

See Strengths And Weaknesses

**Relation To Broader Scientific Literature:**

Yes.

**Theoretical Claims:**

Yes.

---

> ### Author Rebuttal · Authors · 2025-04-01
>
> Dear Reviewer vQva,
>
> Thank you sincerely for taking the time to review our work. We’re encouraged that you found the applications of our method interesting. We appreciate your thoughtful feedback about accessibility for a broader ML audience. We will incorporate a concise, ML-focused framing in the introduction that situates our work within well-known machine learning paradigms such as active learning and surrogate modeling.
>
> Here is the content:
>
> ---
>
> We frame the discovery of optimal gene or drug combinations as a machine learning problem of active search over a high-dimensional combinatorial space, where evaluating each combination (via experiment) is costly. Our method trains a neural surrogate model that predicts the effects of unseen perturbation pairs by combining overparameterized encodings of single-gene outcomes with a learned gene embedding space that models interaction effects. Crucially, the model’s capacity is dynamically scaled with the amount of training data. The surrogate guides new experiment selection via acquisition strategies inspired by Bayesian optimization, with the ability to leverage both exploitation (via maximum predicted effect) and exploration (via ensemble-based uncertainty).
>
> While this work is motivated by biological discovery, similar challenges arise across machine learning domains—including data augmentation policy search in vision [1], cold-start item selection in recommender systems [2], and sample-efficient policy learning in robotics [3]—all of which involve large discrete spaces, costly evaluations, and the need for adaptive modeling and decision making. In such domains, discrete components (e.g., transformations, items, actions) play a role analogous to gene or drug perturbations in our framework: each is embedded in a latent space, and combinations of these embeddings are used to represent and evaluate complex configurations. Our framework shows how these components can be actively selected via a data-adaptive surrogate to enable efficient, scalable discovery.
>
> [1] Cubuk, Ekin D., et al. "Autoaugment: Learning augmentation policies from data." arXiv (2018).
>
> [2] De Pessemier, et al. "Batch versus sequential active learning for recommender systems." arXiv (2022).
>
> [3] Anwar, Abrar, et al. "Efficient Evaluation of Multi-Task Robot Policies With Active Experiment Selection." arXiv (2025).
>
> ---
>
> Additionally, we note that Reviewer kwgr regarded our methodology and framing as well-motivated, and our evidence as convincing. We have carefully considered the suggestions from both Reviewers kwgr and PDeb, and have expanded our experiments accordingly. These results are included in our rebuttal and will be incorporated into the final version of the paper.
>
> Specifically, we have:
>
> (1) further isolated the contributions of the surrogate model and the MPE acquisition strategy
>
> (2) performed a more comprehensive benchmark analysis across various models in active learning iteration
>
> (3) performed hyperparameter tuning for the UCB acquisition function
>
> (4) evaluated the impact of the ensemble size on the performance of NAIAD.
>
>
> We hope this added context addresses your concerns and helps clarify how our contributions relate to mainstream machine learning. Please let us know if you have any further questions. We will be actively available until the end of the rebuttal period. If you feel your concerns are addressed, please consider reevaluating our work. Looking forward to hearing from you!

---

### Decision · Program_Chairs · 2025-05-01

**Decision:**

Accept (poster)

**Comment:**

This paper introduces NAIAD, an active learning framework for discovering optimal gene combinations using a neural surrogate model, adaptive gene embeddings, and a maximum predicted effect acquisition strategy. The work is motivated by the need for efficient exploration of large combinatorial perturbation spaces in gene and drug studies. The authors frame the task within established machine learning paradigms such as surrogate modeling and active acquisition, and they provide comprehensive empirical validation across multiple datasets.

The reviewers offer mixed but mostly positive evaluations:

- Reviewer kwgr gives a strong accept and highlights both the novelty and the empirical strength of the approach, especially the dynamic scaling of embedding dimensionality during training and the simplicity and effectiveness of the MPE acquisition function.

- Reviewer PDeb raises several technical questions in the initial review but acknowledges that the authors addressed them thoroughly in the rebuttal, including with new experiments.

- Reviewer vQva indicates limited familiarity with the topic and did not substantively engage during the discussion period, which limits the weight of that review.

The main contributions of the paper are empirical and systems-oriented rather than theoretical. The combination of adaptive embeddings and greedy acquisition is conceptually straightforward but is implemented thoughtfully and validated convincingly. The authors demonstrate that their approach outperforms existing baselines, and the empirical analysis is thorough, including ablation studies and robustness checks.